# A computational modeling of pri-miRNA expression

**Hansi Zheng[1], Saidi Wang[1], Xiaoman Li[2]\*, Haiyan Hu[1]\***

**1** Department of Computer Science, University of Central Florida, Orlando, Florida, United States of America,
**2** Burnett School of Biomedical Science, College of Medicine, University of Central Florida, Orlando, Florida, United States of America

\* xiaoman@mail.ucf.edu (XL); haihu@cs.ucf.edu (HH)

## Abstract

MicroRNAs (miRNAs) play crucial roles in gene regulation. Most studies focus on mature miRNAs, which leaves many unknowns about primary miRNAs (pri-miRNAs). To fill the gap, we attempted to model the expression of pri-miRNAs in 1829 primary cell types, cell lines, and tissues in this study. We demonstrated that the expression of pri-miRNAs can be modeled well by the expression of specific sets of mRNAs, which we termed their associated mRNAs. These associated mRNAs differ from their corresponding target mRNAs and are enriched with specific functions. Most associated mRNAs of a miRNA are shared across conditions, while on average, about one-fifth of the associated mRNAs are condition-specific. Our study shed new light on understanding miRNA biogenesis and general gene transcriptional regulation.

## Introduction

It is important to study the expression of primary microRNAs (pri-miRNAs). MiRNAs are short endogenous non-coding RNAs. Their biogenesis starts from the transcription of pri-miRNAs, which are processed into precursor miRNAs and eventually become mature miRNAs of ~22 nucleotides (nt) long [1, 2]. Mature miRNAs, usually referred to as miRNAs, bind to their target mRNAs to regulate the target gene expression by degrading the target mRNAs or preventing them from being translated into proteins. The pri-miRNAs are thus the first product of the miRNA biogenesis, which affect the production of the mature miRNAs and the activity of the majority of protein-coding genes the mature miRNAs regulate. Moreover, the expression of pri-miRNA is known to be quite different from that of the corresponding mature miRNAs due to several steps in miRNA biogenesis that process pri-miRNAs to mature miR-NAs [3–7]. In fact, previous studies showed that the expression of mature miRNAs did not correlate well with that of precursor miRNAs, and even the mature miRNAs generated from the same primary miRNA transcript had different expression levels [3, 4]. To understand miRNA-involved gene regulation, it is thus indispensable to study the expression of pri-miRNAs.

Most studies focus on mature miRNAs [1, 8–21]. The expression of mature miRNAs is routinely measured by small RNA-seq experiments [12]. The co-measurement strategy, which

**Data Availability Statement:** We obtained the CAGE data in 1829 samples from https://fantom.gsc.riken.jp/5/datafiles/latest/extra/CAGE_peaks/hg19.cage_peak_phase1and2combined_tpm_ann.osc.txt.gz. The mature miRNA expression data in 378 samples is from the "human.srna.cpm" file at

https://fantom.gsc.riken.jp/5/suppl/De_Rie_et_al_2017/. The 195 pri-miRNA expression in the 1829 samples is in S1 Table (https://doi.org/10.6084/m9.figshare.21578847.v4). The 2312 mRNA expression in the 1829 samples is in S2 Table (https://doi.org/10.6084/m9.figshare.21578847.v4). The LASSO model trained with 1829 samples is available at https://doi.org/10.6084/m9.figshare.21578847.v4.

**Funding:** This work was supported by US National Science Foundation (1661414, 2015838, 2120907). The funder had no role in the design, analysis and publication of this work.

**Competing interests:** The authors have declared that no competing interests exist.

profiles the expression of both mRNAs and miRNAs under related conditions, is commonly employed to pinpoint miRNA target genes [9, 22, 23]. The large-scale analysis of miRNAs and mRNAs gene expression is widely used to define miRNA targets, miRNA modules, miRNA-mRNA co-expression networks, miRNA-transcription factor regulatory networks, etc. [10–12, 15, 22–28]. These studies show that mature miRNA expression often correlates well with their target mRNAs expression under specific conditions.

Despite many studies on miRNAs, the study of pri-miRNA expression is scarce. This scarcity is largely because rarely are the pri-miRNAs known, especially their transcriptional start sites (TSSs) [29]. The TSS of a miRNA gene can be tens or hundreds of thousand base pairs away from the location of the precursor and mature miRNAs [29–33]. Despite over a decade of computational and experimental identification of pri-miRNA TSSs and several collections of pri-miRNA TSS annotation through high-throughput experimental studies, the annotated pri-miRNA TSSs were not consistent across studies [29, 31, 32]. To make it even more challenging, pri-miRNAs usually have low and condition-specific expression, short life span, alternative TSSs under different physiological conditions, etc. Thus, it is no wonder that only a few studies have profiled primary miRNA gene expression so far, and it is unclear whether the profiled expression is truly the pri-miRNA expression [4, 31, 32].

To fill this gap in miRNA studies, we studied the expression of pri-miRNA in 1829 samples measured by the Cap Analysis of Gene Expression (CAGE) experiments [34]. We modeled pri-miRNA expression with the CAGE data, since a CAGE experiment can measure the expression level of pri-miRNAs and mRNAs simultaneously, which alleviates the experimental noises from different experiments. Because pri-miRNA TSSs are largely unknown and inconsistent between different experimental and computational studies, we focus on ~330 pri-miRNA TSSs that are consistent in at least four of fourteen studies [29, 32]. We found that the expression of a pri-miRNA could reliably be modeled by the expression of a set of mRNAs. This set of mRNAs, which we termed the associated mRNAs of this pri-miRNA, were not the target genes of its mature miRNA. For a pri-miRNA, its associated mRNAs were mostly conserved across samples, although a small fraction was condition-specific. Our study shed new light on the expression of pri-miRNAs.

## Material and methods

### 1829 CAGE samples for expression analysis

We downloaded the gene expression data measured by CAGE experiments in 1829 primary cell types, cell lines, and tissues from the FANTOM 5 project [34, 35] (https://fantom.gsc.riken.jp/5/datafiles/latest/extra/CAGE_peaks/hg19.cage_peak_phase1and2combined_tpm_ann.osc.txt.gz). We considered CAGE data instead of other TSS-seq data because of the large number of CAGE samples from the same study, which avoided unexpected discrepancies among samples from different labs. Moreover, the CAGE data can measure the expression level of pri-miRNAs and mRNAs in the same experiments, which alleviates the experimental noise in comparing expression data from different experiments. These CAGE data were normalized by previous studies [34, 35]. To determine the expression level of a pri-miRNA or mRNA, we used the normalized CAGE expression from all peaks located in the neighborhood of the corresponding TSS region. When multiple CAGE peaks occurred in a TSS region, we used the expression of the peak with the maximum expression value to represent the expression value of this TSS region. Alternatively, we tried to use the sum or average of the expression of multiple peaks in a TSS region, which gave a similar model and expression prediction.

## The robust set of miRNA TSSs

We previously collected the annotated pri-miRNA TSSs of 330 miRNAs from fourteen studies [29]. These TSSs were consistently annotated in at least four of the fourteen studies. Since one miRNA may have multiple annotated TSSs, we considered each annotated TSS as a different miRNA and thus considered 369 miRNAs. We then calculated the expression of the 369 miR-NAs in the above 1829 CAGE samples. We filtered miRNAs with zero expression in more than 80% of the CAGE samples, since these miRNAs were not active in the majority of samples. In this way, 195 miRNAs remained in our analysis (S1 Table). Among these miRNAs, 69 miRNAs were annotated consistently between miRBase and miRGeneDB [36, 37].

## The consistent set of mRNAs

Since we model pri-miRNA gene expression with mRNA expression in CAGE samples, we hope that the expression of pri-miRNAs we model in CAGE experiments can approximate the expression we normally observe in the corresponding RNA-seq experiments. In other words, the expression of mRNAs that can be used to model pri-miRNA expression in CAGE samples must be consistent between RNA-seq experiments and the CAGE experiments.

To define such a consistent set of mRNAs to model the pri-miRNA expression in CAGE experiments, we used the 22 tissue samples where both RNA-seq and CAGE data were generated (https://www.ebi.ac.uk/arrayexpress/experiments/E-MTAB-1733/samples/?s_page=1&s_pagesize=500) [38]. The CAGE data and the RNA-seq were processed as previously [38] to obtain the gene expression levels for all 27,493 GENCODE annotated mRNA transcripts. We then calculated Spearman's correlation coefficient for all transcripts across the 22 samples. The transcripts with a correlation larger than 0.75 were chosen as the consistent mRNAs (p-value<2.92E-5, False discovery rate (FDR) <1). For a gene with multiple consistent transcripts, we chose the transcript with the largest correlation to represent this gene and filtered all other transcripts. In this way, we obtained 2312 mRNA transcripts (S2 Table). These mRNAs, together with their annotated TSSs, were used to model the expression of pri-miRNAs.

## 378 samples with both CAGE data and small RNA-seq data

A previous study measured mature miRNA expression with small RNA-seq experiments and pri-miRNA expression with CAGE experiments in 399 of the above 1829 samples [32]. We managed to identify 378 of these 399 samples based on the FANTOM 5 sample ID [34, 35]. By further manual examination, we could not identify additional samples. We thus focused on these 378 samples. The aforementioned study claimed that the expression of pri-miRNAs correlates well with that of mature miRNAs [32], which contradicts the conclusions in several previous studies [3–7]. We thus investigated how different the expression of mature miRNAs was from that of pri-miRNAs in these 378 samples. We downloaded the expression of the mature miRNAs from https://fantom.gsc.riken.jp/5/suppl/De_Rie_et_al_2017/ (S3 Table) and obtained the pri-miRNA expression in these 378 samples as described above (S1 Table). Here we considered all 175 of the above 195 pri-miRNAs with non-zero gene expression in more than 80% of the 378 samples [29].

## The least absolute shrinkage and selection operator (LASSO) regression

We model the pri-miRNA expression by LASSO. LASSO is widely used to model gene expression and select variables previously [39–42]. We use the LASSO tool from the scikit-learn

package (https://scikit-learn.org/0.24/, version 0.24.2). The goal of LASSO is to minimize:

$$\sum_{i=1}^{n}(y_i - \sum_{j=1}^{m}x_{ij}w_j)^2 + \alpha\sum_{j=1}^{m}|w_j|$$

Here the expression of pri-miRNAs in the *i-th* sample was considered as the dependent variable $y_i$, which was a vector of 195 dimensions (S1 Table). The expression of the *j-th* mRNA in the *i-th* sample was considered as the independent variables $x_{ij}$ (S2 Table). The $w_j$ was the coefficient vector of 195 dimensions to describe the importance of the *j-th* mRNA to the pri-miR-NAs. The LASSO method tried to minimize the above function by making certain $w_j$ to be zero and by choosing the remaining mRNAs as the associated mRNAs. Both the values of the dependent and independent variables were normalized to have a mean zero and standard deviation one before the LASSO regression was applied. Note that for seventeen pri-miRNAs, their TSSs were within 100 base pairs of the TSSs of one and only one of the above 2312 mRNAs. We removed these close mRNAs before training the LASSO model to predict the expression of each of these pri-miRNAs.

We considered three neighborhood sizes, 100, 300, and 500 base pairs, around each TSS to measure the expression of the corresponding pri-miRNA or mRNA. For a given neighborhood size, the expression of a pri-miRNA or mRNA was calculated as the normalized expression of the CAGE peak located in the corresponding neighborhood. If multiple CAGE peaks were in the neighborhood of a TSS, we used the largest expression value of these peaks after testing several alternatives and achieving similar model performance. We then applied the LASSO regression to the data for a given neighborhood size. Because the neighborhood size did not affect the model much, we presented the results from the neighborhood size of 100 base pairs.

To measure how well the expression of pri-miRNAs was modeled, we calculated the correlation coefficient of the predicted expression of a pri-miRNA with its actual expression. We calculated both Pearson's correlation and Spearman's correlation per miRNA and per sample. For the per miRNA correlation, we considered the two vectors of expression values across 1829 samples for a miRNA. For the per sample correlation, we considered the two vectors of expression values across 195 miRNAs for a sample. The significance of a correlation *r* was approximated by the t-test p-values $t = \frac{r\sqrt{n-2}}{\sqrt{1-r^2}}$, which asymptotically follows a t-distribution with the degree of freedom of *n-2*, with *n = 195* for the per-sample correlations and *n = 1829* for the per-gene correlations.

## The GO analysis

We inferred the enriched GO terms for the associated mRNAs of every miRNA. The gene symbols of the 2312 consistent mRNAs were considered the population of all genes for this enrichment analysis. We then searched for the enriched GO terms in the associated genes using the GOrilla tool [43]. We recorded the number of GO terms identified by GOrilla with the FDR cutoff 0.1 for each miRNA (S4 Table).

## The calculation of FDR

We calculated FDR in this study with the standard Benjamini Hochberg algorithm [44]. For instance, when we selected pri-miRNAs that had their expression significantly correlated with their mature miRNA expression under the cutoff of FDR 0.01, first, we calculated the p-values of the expression correlation based on the aforementioned t-distribution for each pri-miRNA. Next, we ranked pri-miRNAs with the calculated p-values, from the smallest one to the largest one. Finally, we found the smallest i so that the sum of the p-values from the first pri-miRNA

to the i-th pri-miRNA was no smaller than 0.01, and reported all pri-miRNAs ranked before the i-th pri-miRNA as the significantly correlated pri-miRNAs.

## Results

### The expression of the pri-miRNAs was reliably predicted

We modeled the expression of pri-miRNA in 1829 primary cell types and tissues (Material and Methods). In brief, we considered the TSSs of a miRNA consistently annotated in at least four of fourteen previous studies as the TSS of this pri-miRNA [29]. We then measured the expression of this miRNA by the log normalized counts of reads mapped to the neighborhood of the TSS in the CAGE experiments [34]. Finally, we modeled the expression of the pri-miRNAs with the expression of mRNAs in these 1829 CAGE experiments by the LASSO regression [45]. LASSO selected a subset of mRNAs for each pri-miRNA. We called these selected mRNAs for a pri-miRNA as its associated mRNAs (S5 Table).

We found that the expression of the associated mRNAs could reliably model the expression of pri-miRNAs. We calculated the correlation of the predicted pri-miRNA expression value by the LASSO model with the true pri-miRNA expression value measured by CAGE. The minimum, mean, and median Pearson's correlation per miRNA was 0.79, 0.91 and 0.92, respectively (p-value = 0 for all correlations, S6 Table). Similar, the minimum, mean and median Spearman's correlation per miRNA was 0.27, 0.82 and 0.83, respectively (p-value = 0 for all correlations, S6 Table). If we measured the similarity of the predicted expression in every sample, the minimum, mean, and median Pearson's correlation per sample was 0.48, 0.88 and 0.90, respectively (p-value <1.79e-13 for all correlations, S7 Table). Correspondingly, the minimum, mean and median Spearman's correlation per sample was 0.42, 0.84 and 0.86, respectively (p-value < 4.91e-10 for all correlations, S7 Table). The significant correlation suggested that the expression of the associated mRNAs could reliably model the pri-miRNA expression.

We further examined the miRNAs with their expression accurately predicted (Pearson's correlation >0.90) and the miRNAs with the expression not predicted so well (Pearson's correlation < = 0.90) (Table 1). Note that the correlation was larger than 0.79 for all miRNAs. We found that the miRNAs with their expression accurately predicted were pri-miRNAs with much higher expression levels and much larger expression variation. On the contrary, miRNAs that were not modeled so well were pri-miRNAs with low expression and low expression variation. For instance, the miRNAs modeled well had a median expression value and a standard deviation of 22.31 and 62.57, while the miRNAs modeled not so well had the corresponding value as 8.80 and 14.64, respectively. In fact, for every miRNA modeled not so well, they had zero expression in at least 82.39% of the samples. In other words, these miRNAs were not modeled so well because they were not so related to the experimental conditions these samples considered. If we excluded the miRNAs with their expression standard deviation smaller than 3 in these 1829 samples, the Pearson's correlation of the predicted expression with the true

**Table 1. The minimum, maximum, and median of Pearson's correlation coefficient of three groups of miRNAs.**

| Group | Min correlation | Max correlation | Median correlation | Min expression | Max expression | Median expression |
|---|---|---|---|---|---|---|
| High | 0.90 | 1.00 | 0.94 | 2.67 | 313.48 | 62.57 |
| Low | 0.79 | 0.90 | 0.87 | 1.45 | 96.21 | 8.80 |
| Active | 0.83 | 1.00 | 0.92 | 2.26 | 313.48 | 18.39 |

The miRNAs in the high group had a correlation > 0.9. The miRNAs in the low group had a correlation < = 0.9. The miRNAs in the active group had the standard deviation of expression larger than three.

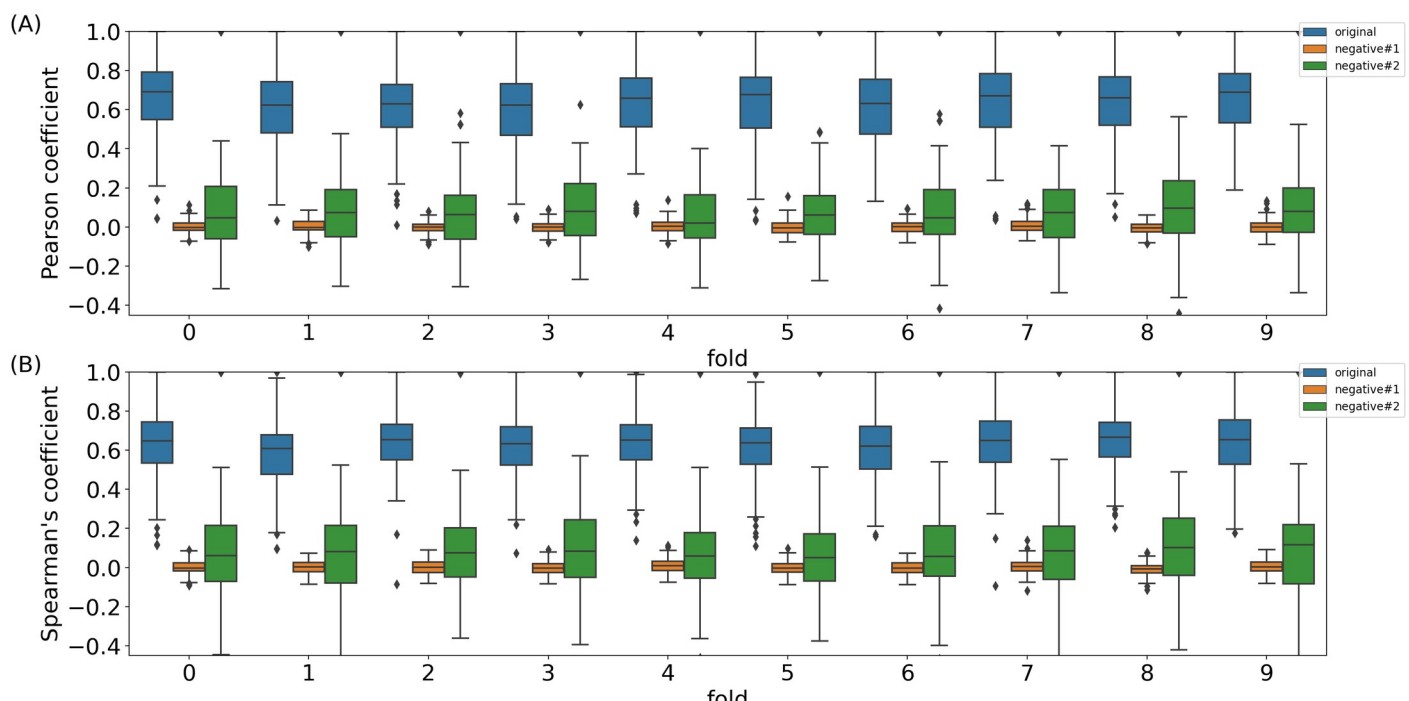

**Fig 1.** The box plot of (A). Pearson's and (B). Spearman's correlation coefficient of the predicted and true expression of pri-miRNAs in ten two-fold cross-validation experiments. For each experiment, the three boxes in order correspond to the original data, the first negative control dataset and the second negative control dataset.

expression was larger than 0.83 for each of the remaining 188 miRNAs (p-value = 0). The good modeling of the expression of the 188 pri-miRNAs suggested that the expression of almost all pri-miRNAs could be predicted well by their associated mRNAs with enough samples.

The above analysis was based on all 1829 samples. We next did ten experiments with two-fold validation to see whether the LASSO models trained on a subset of randomly selected samples could predict pri-miRNA expression in the remaining samples (Fig 1). In each experiment, we randomly separated all 1829 samples into two groups, and then made sure to remove testing samples that were from the same primary cell types, tissues or cell lines used in training. On average, the minimum, mean and median Pearson's correlation per miRNA was 0.10, 0.62 and 0.64, respectively. The minimum, mean and median Spearman's correlation per miRNA was 0.14, 0.61 and 0.63, respectively. Correspondingly, the minimum, mean and median Pearson's correlation per sample was -0.2, 0.61, and 0.65, respectively. The minimum, mean and median Spearman's correlation per sample was -0.17, 0.59 and 0.63, respectively. Although the correlation was much smaller than the model trained with all 1829 samples, it was still significantly large (p-value of the mean and median correlation was 0), suggesting that the pri-miRNA expression was reliably predicted. The lower correlation in the cross-validations also indicated that the regulation of the pri-miRNA expression was sample-specific, and the model inferred from a specific subset of samples would predict pri-miRNA expression in the remaining samples not so well as the model trained on these remaining samples.

To further justify the significant correlation between the predicted and actual expression of pri-miRNAs, we compared the above correlation with the correlation of the predicted and actual expression of pri-miRNAs on two negative control datasets. In each negative dataset, we kept the original pri-miRNA expression in the 1829 samples while randomly permuted the expression of mRNAs in these samples. For the first negative control dataset, we randomly permuted the expression values of each mRNA across the 1829 samples. For the second negative

control dataset, we randomly permuted the expression values of mRNAs within each of the 1829 samples. We then trained the LASSO models to predict pri-miRNA expression on each negative control dataset as above. We found that the trained models on the negative control datasets could not predict pri-miRNA expression well. For instance, in the corresponding two-fold cross-validation experiments, the correlation from the negative control datasets was much smaller than that from the original 1829 samples we presented above (Fig 1).

## The associated mRNAs were different from target mRNAs

Previous studies correlated mature miRNA expression with the expression of their target mRNAs [11, 22]. The target mRNAs of a miRNA contain its target sites and can be bound by this miRNA. We thus compared the associated mRNAs inferred above with the target mRNAs for each miRNA. The target mRNAs of a miRNA were retrieved from two sources (Material and Methods). One was the targets computationally predicted by the TargetScan Version 7.2 tool [8]. The other was the miRNA targets experimentally validated in the miRTarBase Version 8.0 database [46].

We found that the associated miRNAs differed from the target mRNAs for every miRNA. On average, only 32.58% of the associated mRNAs were the TargetScan mRNAs, while about 0.89% of the TargetScan mRNAs were the associated mRNAs for a miRNA. Similarly, the corresponding percentage was 2.24% and 8.05%, respectively, for the miRTarBase mRNAs. The overlap between the two types of mRNAs suggests the difference between the two types of mRNAs and the fact that the target mRNAs do not have a correlated expression with their pri-miRNAs in general.

Since the associated mRNAs modeled the pri-miRNA expression better than the target mRNAs, we hypothesized that their expression correlated better with the expression of the corresponding pri-miRNA than the target mRNAs. We found that for a miRNA, the expression of most of its associated mRNAs indeed correlated better with its expression than the expression of its target mRNAs across the 1829 samples, no matter whether the target mRNAs were defined by targetScan or miRTarBase (Fig 2). When we compared the expression correlation of the associated mRNAs with the expression correlation of the target mRNAs, we found that 98.46% of miRNAs had a significantly higher correlation with their associated mRNAs than their target mRNAs (Mann-Whitney p-value < 0.001).

To evaluate whether the miRNA target genes are enriched in the associated genes, we performed the hypergeometric test for miRNA target genes from TargetScan and miRTarBase separately (S8 Table). In brief, for a given pri-miRNA, assume it has $n$ associated mRNAs, $m$ of which are its target genes. Assume there are $M$ of its target genes in the $N = 2312$ consistent genes we considered above. Then the p-value of observing at least $m$ target genes in its associated mRNAs is calculated by the hypergeometric testing as $\sum_{k=m}^{n} \frac{C(N-M,n-k) \times C(M,k)}{C(N,n)}$, where $C(x,y) = \frac{x!}{y!(x-y)!}$ for any non-negative integers $x$ and $y$. We found that 192 and 178 of the 195 miRNAs had a hypergeometric testing p-value > 0.01 for TargetScan and mirTarBase targets, respectively (10% percentile of the p-values 0.052 and 0.085, respectively, S8 Table). The large enrichment p-values showed that the miRNA targets are usually not enriched in the associated mRNAs for almost all miRNAs we tested. Note that the pri-miRNAs with higher expression and larger expression variation did not have smaller miRNA target enrichment p-values than other pri-miRNAs.

## The expression of the majority of mature miRNAs do not correlate well with the expression of pri-miRNAs

The difference between the associated mRNAs and the target mRNAs suggests that the expression of pri-miRNAs is different from that of mature miRNAs. Otherwise, since mature miRNAs have a correlated expression pattern with their target mRNAs under specific conditions

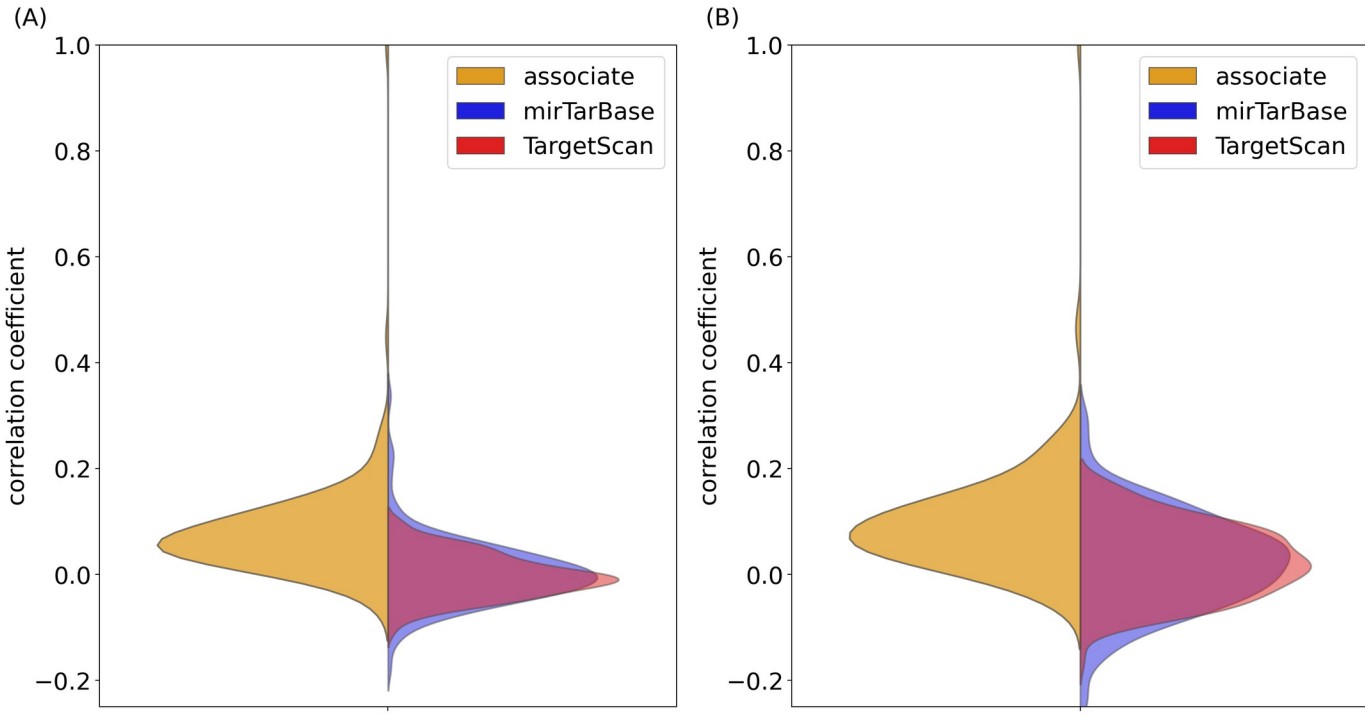

**Fig 2. The Pearson's and Spearman's correlation of pri-miRNAs and their target genes from miTarBase and TargetScan compared with the correlation of pri-miRNAs and their associated genes.**

[22, 26], we would have seen more target mRNAs included in the associated mRNAs. This implication is supported by several previous studies [3–7] while contradicting another study [32]. We thus studied the miRNA expression in the 378 samples with CAGE data and small RNA-seq data, which we found were used by this contradicting study.

We found that the expression of pri-miRNAs has a relatively low correlation coefficient with that of mature miRNAs for at least 101 (57.71%) of the 175 miRNAs we studied (S9 Table, FDR < 0.01). This percentage was based on the Pearson's correlation coefficient of the 175 miRNAs in the 378 samples and a FDR cutoff of 0.01. Since one pri-miRNA may correspond to multiple mature miRNAs, we chose the largest correlation between a pri-miRNA and its mature miRNAs as the correlation here. The Pearson's correlation coefficient ranged from 0 to 0.87, with its 57.71 percentile as 0.1592 (FDR<0.01), suggesting that a large fraction of pri-miRNAs have a low correlated expression with their mature miRNAs. Note that the calculated Pearson's correlation coefficient here was highly similar to that in the contradicting study [32]. We made a different conclusion because we were considering individual miRNAs while the other study was considering all miRNAs together. The Spearman's correlation coefficient gave a similar but smaller percentage, likely due to its less informative nature with only the ranks instead of the actual values. We also noticed that the pri-miRNAs with higher expression and larger expression variation do not have a larger expression correlation with their mature miRNAs than other pri-miRNAs (S10 Table, two-sided Mann-Whitney p-value = 0.9828).

The contradicting study measured the expression of pri-miRNAs with their own TSSs, which were different from what we used here (S3 Table). We thus also repeated the above analysis with their defined TSSs. This previous study defined TSSs and mature miRNA for 175

miRNAs we studied here. We found that 102 (58.28%) of the 175 miRNAs had a low correlated expression between the pri-miRNAs and the mature miRNAs (FDR<0.01). Interestingly, 101 of the 102 miRNAs were also included in the above 101 miRNAs.

## The associated mRNAs were biologically sound

The above analysis showed that the associated mRNAs (S5 Table) had a more correlated expression pattern with the pri-miRNAs than the target mRNAs, implying the functionality of these associated mRNAs for each miRNA. We further investigated other properties of the associated mRNAs and found that most of them were not affected by the neighbor sizes around miRNA TSSs and different samples used. Moreover, these associated mRNAs for most miRNAs had enriched functional annotations.

We checked how different neighborhood sizes might affect the associated mRNAs inferred for a miRNA (Material and Methods). We considered 100, 300 and 500 base pairs around an annotated miRNA TSS as the TSS regions to measure the normalized gene expression of miRNAs and mRNAs. For the three neighborhood sizes, the median number of associated mRNAs identified was 172, 171, and 169, respectively. We also studied the correlation of the predicted and true expression of each pri-miRNA with respect to different TSS neighborhood sizes and found that the neighborhood sizes had no significant effect on predicting miRNA expression (Table 2). On average, 71.74% of the associated mRNAs were the same for a pri-miRNA when different neighborhood sizes were used, suggesting that the associated mRNAs were likely to be biologically meaningful and intrinsically related to the corresponding miRNAs.

We also studied how different samples may change the associated mRNAs inferred for a miRNA. For a given miRNA, we compared its associated mRNAs inferred from each fold in the ten experiments of two-fold cross-validation with the associated mRNAs inferred from all samples (Fig 3A). Here the neighborhood size was set to be 100 base pairs since different neighborhood sizes did not change the predictions much. Interestingly, the median number of the associated mRNAs was slightly fewer, around 140. Moreover, on average, about 50.09% of the associated mRNAs were shared between every fold and the model from all samples, suggesting that about half of the associated mRNAs are condition-specific. We also found that 33.20% of the associated genes were shared by at least five experiments (Fig 3B). Together with the above analysis indicated that the subset of samples used will affect how model behave, potentially due to tissue-specific expression of pri-miRNAs and mRNAs.

With the associated mRNAs for each miRNA, we investigated whether they significantly shared gene ontology (GO) functions [47]. With the FDR cutoff 0.1, 10.76% pri-miRNAs indeed had at least one GO term significantly shared by its associated mRNAs. Among these pri-miRNAs, the median number of significantly shared GO terms was three (S4 Table). The enriched GO terms are usually consistent with the function of the miRNAs in literature. For instance, several GO terms including "regulation of cellular biosynthetic process" (GO:0031326, p-value = 1.75E-5, FDR Q-value = 8.21E-2), "regulation of cellular macromolecule biosynthetic process" (GO:2000112, p-value = 3.11E-5, FDR Q-value = 9.69E-2) and "regulation of macromolecule biosynthetic process" (GO:0010556, p-value = 3.44E-5, FDR Q-

**Table 2. The minimum, maximum, mean, and median of Pearson's correlation coefficient of three different TSS neighborhood size of miRNAs.**

| TSS neighborhood size | min | max | mean | median |
|---|---|---|---|---|
| 100 | 0.79 | 1.00 | 0.92 | 0.92 |
| 300 | 0.78 | 1.00 | 0.92 | 0.92 |
| 500 | 0.82 | 1.00 | 0.92 | 0.92 |

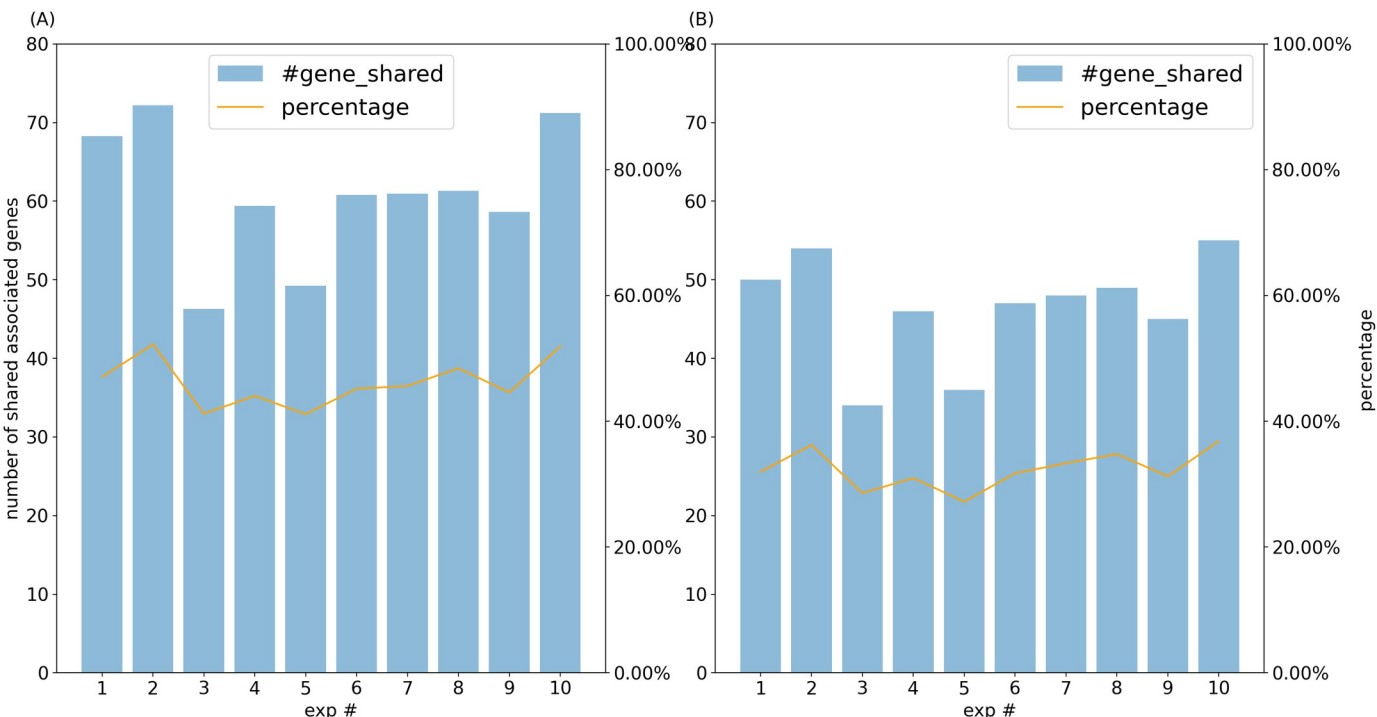

**Fig 3.** (A) The number and percentage of the associated mRNAs in each experiment shared by the full model with all samples. (B) The number and percentage of the associated mRNAs in each experiment shared by at least four other experiments.

value = 8.05E-2,) were enriched in the associate mRNAs of hsa-mir-92a, including the associate genes e74-like factor 4 (ELF4) and the smad family member 6 (SMAD6) (Fig 4). The hasmir-92a represses the viability and migration of nerve cells in Hirschsprung's disease by regulating the KLF4/PI3K/AKT pathway [48]. ELF4 is a the transcription factor that controls proliferation and homing of CD8+ T cells via the KLF4 and KLF2 [49]. hsa-mir-92a also inhibits SMAD6-mediated RUNX2 degradation and promotes osteogenic differentiation of BMSCs [50]. These enriched GO terms support the implied function of hsa-mir-92a and the biological significance of the inferred associated mRNAs of hsa-mir-92a. Note that not all miRNAs had their associated mRNAs significantly shared GO functions, partially due to the imperfect GO annotation.

## Discussion

The study of pri-miRNA expression is still in its infancy. Here we modeled the pri-miRNA expression in 1829 primary cells and tissues. We demonstrated for the first time that the expression of the associated mRNAs could reliably predict the expression of the pri-miRNAs. These associated mRNAs are different from their target mRNAs while having a more correlated expression with the pri-miRNAs than the target mRNAs. For most miRNAs, their associated mRNAs significantly shared GO functions. The above observations were valid for miRNAs defined in miRBase and miRGeneDB [36, 37] (S1–S5 Tables). Our study may thus provide a new way to indirectly measure the expression of pri-miRNAs under different experimental conditions that are challenging to measure directly.

Several studies showed that the pri-miRNA expression is quite different from the corresponding mature miRNA expression [3–7]. However, another study claimed a good expression correlation of pri-miRNAs and mature miRNAs in 399 samples with both CAGE and

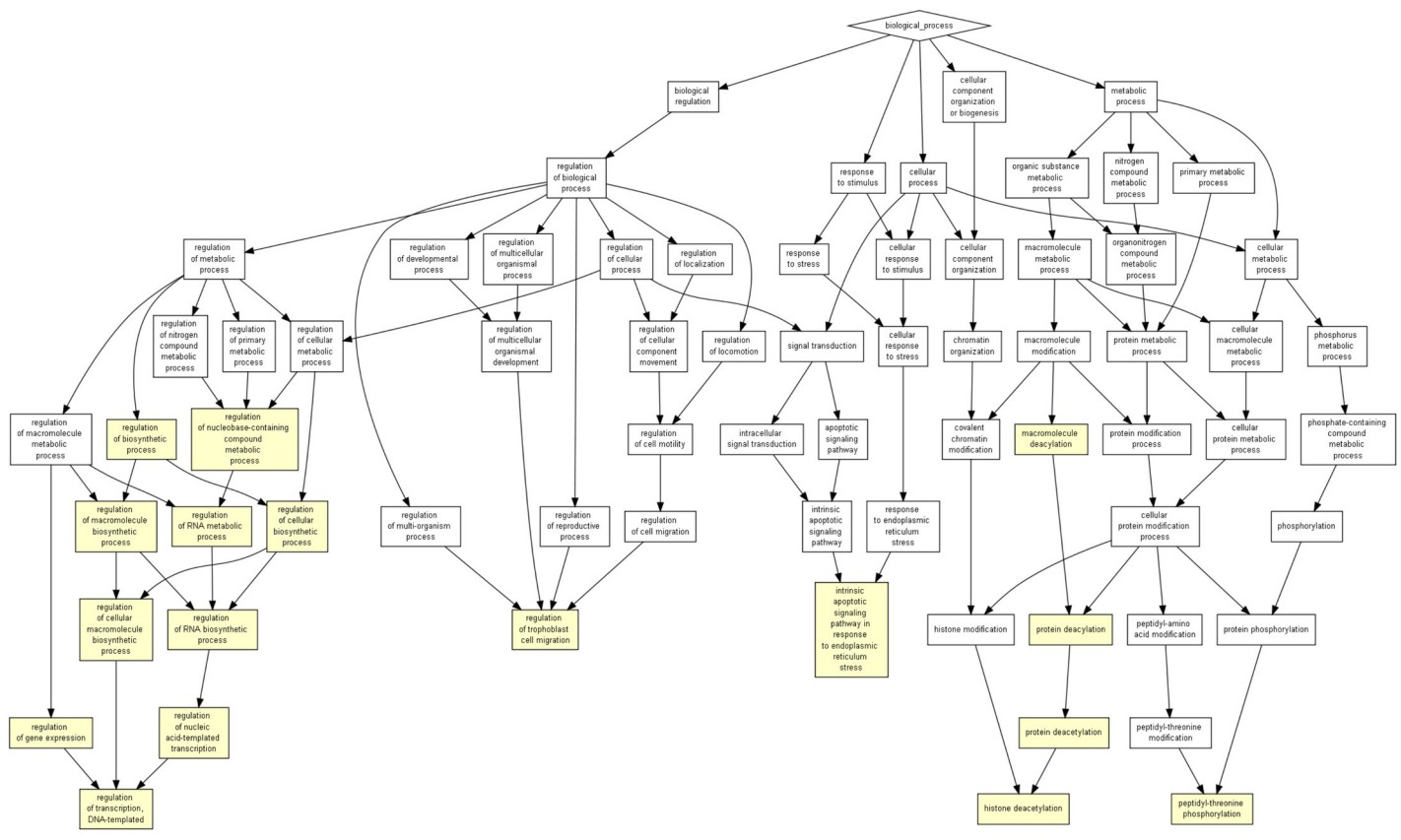

**Fig 4. The enriched biological process GO terms of the associate genes for hsa-miR-92a.**

small RNA-seq data. We successfully matched 378 of these 399 samples with 378 of the 1829 samples we used and studied how pri-miRNA expression correlated with mature miRNA expression in these 378 samples (S9 Table). We found that the majority of mature miRNAs have a low correlated expression with their pri-miRNAs (FDR<0.01), even with the original data used in this previous study, suggesting that mature miRNA expression is different from pri-miRNA expression.

We demonstrated that the expression of almost all pri-miRNAs could be reliably modeled. In fact, the predicted expression had a correlation >0.82 with the actual expression for all pri-miRNAs with the standard deviation of the expression larger than three in the 1829 samples (p-value = 0, Table 1). We showed that the miRNAs modeled not so well were likely to have low expression and low expression variation. With more samples available in the future, one could model the expression of more pri-miRNAs.

More than a dozen studies previously predicted or annotated miRNA TSSs [20, 29, 32, 51–53]. These predicted or annotated TSSs were often inconsistent between different studies [29, 32]. A recent survey identified 369 miRNA TSSs consistent in at least four previous studies for 330 miRNAs [29]. We selected 195 of these 369 miRNA TSSs that showed transcriptional activities in at least 80% of the 1829 samples in this study. Although we did not consider the alternative miRNA TSSs, these miRNA TSSs were likely the best set we could have currently since they were shown to have better qualities previously [29, 32].

In addition to the 195 miRNA TSSs, we considered 2312 mRNAs to model the expression of pri-miRNAs. Because we measured gene expression through the CAGE data, we narrowed

our analysis to the 2312 mRNAs with consistent expression patterns in CAGE and RNA-seq experiments. Moreover, we applied LASSO to model the expression of pri-miRNAs, thus more likely to capture only linear relationships between the miRNAs and the mRNAs. In the future, with more accurate annotation of miRNA TSSs and a better understanding of TSS-seq data, more sophisticated approaches and more comprehensive studies can be carried out to involve more pri-miRNAs and more mRNAs [54–56].

## Supporting information

**S1 Table. 195 pri-miRNA expression in 1829 samples.** (tab delimited txt) available at https://doi.org/10.6084/m9.figshare.21578847.v5.
(TXT)

**S2 Table. 2312 highly consistent mRNA expression in 1829 samples.** (tab delimited txt) available at https://doi.org/10.6084/m9.figshare.21578847.v5.
(TXT)

**S3 Table. 175 mature miRNA expression in 378 samples.** (tab delimited txt) available at https://doi.org/10.6084/m9.figshare.21578847.v5.
(TXT)

**S4 Table. The enriched GO term for associate mRNAs of 195 pri-miRNAs.** (tab delimited txt) available at https://doi.org/10.6084/m9.figshare.21578847.v5.
(TXT)

**S5 Table. The associate genes, TargetScan targets, mirTarBase targets for 195 pri-miRNAs.** (tab delimited txt) available at https://doi.org/10.6084/m9.figshare.21578847.v5.
(TXT)

**S6 Table. The Pearson's and spearman's correlation coefficient per pri-miRNA between predicted and true pri-miRNA expression.** (tab delimited txt) available at https://doi.org/10.6084/m9.figshare.21578847.v5.
(TXT)

**S7 Table. The Pearson's and Spearman's correlation coefficient per sample between predicted and true pri-miRNA expression.** (tab delimited txt) available at https://doi.org/10.6084/m9.figshare.21578847.v5.
(TXT)

**S8 Table. The statistics of the hypergeometric testing p-values for target mRNAs enrichment in the associated mRNAs.** (tab delimited txt) available at https://doi.org/10.6084/m9.figshare.21578847.v5.
(TXT)

**S9 Table. The Pearson's and spearman's correlation coefficient between pri-miRNAs and mature miRNAs.** (tab delimited txt) available at https://doi.org/10.6084/m9.figshare.21578847.v5.
(TXT)

**S10 Table. The statistics of the Pearson's correlation coefficient between two groups of pri-miRNAs and their respective mature miRNAs.** (tab delimited txt) available at https://doi.org/10.6084/m9.figshare.21578847.v5.
(TXT)

## Author Contributions

**Data curation:** Hansi Zheng.

**Formal analysis:** Hansi Zheng, Saidi Wang, Xiaoman Li, Haiyan Hu.

**Funding acquisition:** Xiaoman Li, Haiyan Hu.

**Methodology:** Xiaoman Li, Haiyan Hu.

**Supervision:** Xiaoman Li, Haiyan Hu.

**Validation:** Hansi Zheng.

**Visualization:** Hansi Zheng.

**Writing – original draft:** Xiaoman Li, Haiyan Hu.

**Writing – review & editing:** Saidi Wang, Xiaoman Li, Haiyan Hu.

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
