## [Decision Letter · Decision Letter 0]

11 Oct 2022

PONE-D-22-18168A computational modeling of pri-miRNA expressionPLOS ONE

Dear Dr. Li,

Thank you for submitting your manuscript to PLOS ONE. After careful consideration, we feel that it has merit but does not fully meet PLOS ONE’s publication criteria as it currently stands. Therefore, we invite you to submit a revised version of the manuscript that addresses the points raised during the review process.

 Both Reviewers raised several important issues that you are kindly asked to clarify, correct where necessary and solve. Specifically, a correct treatment of redundancy is mandatory to avoid the presence of too similar samples in the training and in the test, as discussed by Reviewer 1. You are also asked to solve the issues related to miRNA nomenclature, raised by Reviewer 2 in his first point,  that can significantly affect part of the results obtained. Also, the quality of the figures needs to be improved (Reviewer 2 in his second point). Furthermore, both Reviewers listed other suggestions that for sure would contribute to produce a better and more interesting manuscript (discussing about the biological significance of the model, discussions about poorly discussed thresholds and so on).

We look forward to receiving your revised manuscript.

Kind regards,

Manuela Helmer-Citterich

Academic Editor

PLOS ONE

Journal Requirements:

This work was supported by the National Science Foundation (1661414, 2015838, 2120907).

However, funding information should not appear in the Acknowledgments section or other areas of your manuscript. We will only publish funding information present in the Funding Statement section of the online submission form. 

This work was supported by US National Science Foundation (1661414, 2015838, 2120907). The funder had no role in the design, analysis and publication of this work.

Reviewers' comments:

Reviewer's Responses to Questions

**Comments to the Author**

1. Is the manuscript technically sound, and do the data support the conclusions?

Reviewer #1: Yes

Reviewer #2: Partly

2. Has the statistical analysis been performed appropriately and rigorously? 

Reviewer #1: Yes

Reviewer #2: No

3. Have the authors made all data underlying the findings in their manuscript fully available?

Reviewer #1: Yes

Reviewer #2: Yes

4. Is the manuscript presented in an intelligible fashion and written in standard English?

Reviewer #1: Yes

Reviewer #2: Yes

5. Review Comments to the Author

Reviewer #1: The authors perform a computational analysis of pri-miRNA expression. They select a set of 2312 mRNAs whose expression correlates between CAGE & RNAseq experiments to model 330 pri-miRNA expression across 1829 samples.

• what would be the biological significance of the model? Are the associated mRNA enriched in biological functions? Which ones? The text is missing a more detailed description of the results from the biological & functional perspective.

• Another explanation why the results are significantly worse in the Cross Validation (CV) WRT the large model is lack of generalization. This should be at least discussed in the text. The lack of training and test should be addressed. Additionally even in CV it is highly recommended to exclude from the training, the samples in which the test predictions are made or very similar samples to the test set that are present in the training.

• How is the performace of the model affected by the choice of the TSS neighborhood? Can the authors give a quantitative estimate of the variation in predicted and observed pri-miRNA correlation compared to the model described?

• How many of the target mRNAs are present in the selected 2312 mRNAs? The authors conclude that the associated mRNAs are different from the target mRNAs, but they should report how many were excluded after their correlation filter and quantify the correlation of associated mRNAs and target mRNAs. Higher correlation of the associated mRNAs WRT target might be due to model overfitting. This should be discussed in the text.

• Related to the previous point, I would suggest an analysis of target genes enrichment across associated miRNAs of pri-miRNAs showing higher expression & larger variation WRT the rest of pri-miRNAs? Also, do these pri-miRNAs correlate better with their respective mature miRNAs?

• Does including genes that show less correlation than the threshold used (r=0.75) between CAGE & RNAseq increase noise in the model? Can the performance change be measured and discussed in the text?

• finally I would encourage the authors to publish their model in a publicly accessible repository.

Reviewer #2: The manuscript by Li and colleagues explores the possibility of using mRNA expression profiles to infer the pri-miRNA expression pattern in human samples. The efforts by the authors to set up a consistent annotation of pri-miRNA TSS and recovering expression values from a large group of samples are noteworthy, and the LASSO analysis seems appropriate to answer the main point. However, there are a number of issues that impair the scientific soundness of the manuscript and make it not suitable for publication.

Major points:

1) In Table S4, the miRNA nomenclature is non standard. the "star" (*) notation to denote passenger strands has been abandoned a long time ago and mature miRNAs are named after the portion (5p, 3p) of the corresponding hairpin they arise from. Furthermore, the targetScan_targets field is "N/A" for ~235 out of 278 miRNAs. However, most of these miRNAs "without targetScan tqargets" have a long list of targets predicted by targetScan. I suspect that automatic retrieval of targetScan targets failed (perhaps due to inappropriate nomenclature of mature miRNAs in the dataset by the authors). This casts a serious shadow on the claim that "These associated mRNAs differ from their

corresponding target mRNAs" as in 85% of the cases the miRNAs targets are (mistakingly) missing, hence they of course do not overlap "associated genes". This part of the analysis should be repeated after fixing the issue with the missing targetScan targets.

2) Figure quality can be dramatically improved in terms of resolution, labeling, clarity of legends, and most importantly, the kind of data reported. In Figure 1 in its current form the only take home message is that correlations assessed by Pearson or Spearman algorithms yield comparable values. However, the point here is how good are the observed correlations (compared to a putative null hypothesis). Therefore, comparing the distribution of correlation values obtained using LASSO modeling with mock data using a boxplot or violinplot would be more appropriate.

It is not entirely clear to me what is depicted in Fig 1A, did the authors compare the "real" values and "computed" values on the very same identical data (i.e. traning data == test data)? If this is the case, such validation is tautological and should be removed. Furthermore, what is the rationale for choosing 10-fold cross validation (over 5-fold cross validation or any other n-fold?). The number of samples (> 1800) is very large, a robust trend should be validated by simply splitting the dataset in two halves. n-fold cross validation is usually more appropriate when dataset size is limited.

In figure 2 only 2 example miRNAs are reported. A cumulative picture reporting the general trend for all miRNAs should be provided instead. Furthermore, mirTarBase and targetscan targets should be analysed separately. Alternatively a boxplot displaying side to side (1) the distribution of the median correlation with "associate genes" for each miRNA (2) the distribution of the median correlation with targetScan target genes for each miRNA (3) the distribution of the median correlation with mirTarBase target genes for each miRNA.

Figure 3

Rather than reporting the observed outcome for each of the 10-fold cross validations, an average (or median) +/- SD should be reported. Howevver, check my comment above about the choice of 10-fold cross validation over other validation strategies available.

3) The cutoff explored by the authors (a value > 0 in at least 20% of samples) for miRNA-TSS CAGE data will yield a zero inflated dataset. This is a relevant limitation of the study. Having a 0 in 80% of cases implies that any algorithm predicting always "0" will attain an 80% success. The authors should consider a more stringent cutoff to avoid such issues, restricting their analysis to the pri-miRNAs which exhibit a value > 0 in 80% of the cases. Otherwise, a strategy to cope with zero inflated data should be outlined and applied. Incidentally, 80% > 0 is the cutoff the authors apply to mature miRNA analysis. One might wonder why two such different cutoffs (20% > 0 for pri-miRNA and 80% > 0 for mature miRNAs have been applied in the same manustript.

4) the claim that data reported in ref (31) are due to an error or misinterpretaion of the data by the authors i not supported by the data. While I totally agree with Li and colleagues as long as the point is that pri-miRNAs and the corresponding mature miRNAs display a rather poor correlation (and I believe this makes pri-miRNA quantification a very poor proxy for mature miRNA expression) in (31) Fig 3C de Rie et al have compared the distribution of correlations observed between CAGE-pri-miRNA data (median ~ 0.2 as assessed by the figure) with the ones observed after breaking ties in the data (randomly permuting CAGE-pri-miRNA data and corresponding mature miRNA data), an accepted method of mimicking the null hypothesis. de Rie et al make a formally correct statement that the amount of correlation they observe is significantly higher than the one expected by chance. Furthermore the numbers they observed match quite closely the ones reported by Li inhere (a 55 percentile of 0.201).

So de Rie et al are correctly reporting a statistically significant correlation between CAGE-pri-miRNA and mature miRNAs, and their data perfectly agree with the analysis reported inhere. If the authors feel that a median correlation of 0.2 (although statistically significant) may be of rather poor interest from a biological point of view, the discussio would be the right session of the manuscript to laborate on that, but a negative comment on de Rie et al in the results section (while reporting essentially the same numbers) to question their interptretation of the data is inappropriate.

I would also state here that I am not in any way associated with any of the authors of (31).

5) The GO terms analysis is not reported, not even in the supplementary data. It would be interesting to see which are the GO terms associated with "associated genes" of each miRNA, whether they match the GO terms associated with their target (e.g. it would be conceivable that a set of regulators or transcription factors involved in apoptosis affect (and are hence associated with) the transcription of a mature miRNA targeting apoptosis related genes). Th authors should provide the full output of their GO terms analysis and elaborate on that.

6) page 14 "or the three neighborhood sizes, the median number of associated mRNAs identified was 164, 163, and 164, respectively, indicating the robustness of the inferred associated mRNAs" This is not correct. The fact that using different neighborhood sizes does not affect the associated mRNAs is rather telling us that the neighborhood window has a poor effect on the assessment of pri-miRNA-CAGE, thus yielding consistent input data (regardless of the neighborhood size) to the same LASSO algorithm, which obviously results into very similar output. However, any conclusion on the robustness of the algorithm itself would be an over-interpretaion of this observation. The robustness of the approach could be validated instead by testing it independently on different datasets (or different slices of the same dataset).

Minor points:

1) In table S4 some miRNAs are repeated twice (e.g miR-101-1, miR-29b-2) and in some cases (miR-101-1) the TSS taken into account are almost completely overlapping.

2) the datasets, miRNAs and samples analysed, although described in the text, should be described in tables depicting the number of samples, miRNAs and pri-miRNA CAGE fed into each analysis. GO terms output should also be provided in tables (and possibly representative examples reported as figures). The validation data about the LASSO approach should also be reported as supplementary tables (i.e. values underlying Fig 1).

3) the authors should elaborate on the biological significance of inferring pri-miRNA expression based on a set of mRNAs. Why this finding is potentially interesting from a biological point of view? do they expect that "associate mRNAs" may encode for transcription factors, RNA binding enzymes or other proteins directly involved in promoting/repressing transcription of pri-miRNAs?

6. PLOS authors have the option to publish the peer review history of their article (what does this mean?). If published, this will include your full peer review and any attached files.

Reviewer #1: **Yes: **Antonio Palmeri

Reviewer #2: No

---

## [Author Response · Author response to Decision Letter 0]

22 Nov 2022

Reviewer #1: 

Comment 1: What would be the biological significance of the model? Are the associated mRNA enriched in biological functions? Which ones? The text is missing a more detailed description of the results from the biological & functional perspective.

Response: Thanks for great questions and suggestions. We emphasized the significance in introduction and discussion. We provided the GO analysis results of miRNAs as an additional supplementary file S4. We also added some further descriptions of the biological significance of the model based on the GO enrichment analysis. 

Comment 2: Another explanation why the results are significantly worse in the Cross Validation (CV) WRT the large model is lack of generalization. This should be at least discussed in the text. The lack of training and test should be addressed. Additionally even in CV it is highly recommended to exclude from the training, the samples in which the test predictions are made or very similar samples to the test set that are present in the training.

Response: Thanks for great comments! We agree that the correlation in the CV is not as high as that in the large model, which implies that the miRNA expression is condition-specific and the model trained from a subset of samples cannot fully explain the expression variation in new samples. We emphasized this condition-specific nature in the modified manuscript. Note that in both CV and the large model, the correlation is statistically significant (p-value 0). We also followed your suggestion to remove similar samples in testing that are used in training. We found that the trained model still can explain the pri-miRNA expression well, although not as well as the full model. 

Comment 3: How is the performance of the model affected by the choice of the TSS neighborhood? Can the authors give a quantitative estimate of the variation in predicted and observed pri-miRNA correlation compared to the model described?

Response: Thanks for the comment! The performance of the model is not affected by the choice of the neighborhood sizes we used. We provided additional information in the first paragraph of page 15.

Comment 4: How many of the target mRNAs are present in the selected 2312 mRNAs? The authors conclude that the associated mRNAs are different from the target mRNAs, but they should report how many were excluded after their correlation filter and quantify the correlation of associated mRNAs and target mRNAs. Higher correlation of the associated mRNAs WRT target might be due to model overfitting. This should be discussed in the text.

Response: Thanks for the great comments! In the modified version, we reported the bynber if target mRNAs in the 2312 mRNAs in Supplementary Tables S5 and S8. We also calculated the hypergeometric testing p values and showed that the target genes are not enriched in the associated genes (Supplementary Table S8). 

Comment 5: Related to the previous point, I would suggest an analysis of target genes enrichment across associated miRNAs of pri-miRNAs showing higher expression & larger variation WRT the rest of pri-miRNAs? Also, do these pri-miRNAs correlate better with their respective mature miRNAs?

Response: Thanks for the suggestion. We added the enrichment analysis as mentioned above. We found no significant difference between the two groups of pri-miRNAs based on enrichment analysis between target mRNAs and associated mRNAs. The pri-miRNAs showing higher expression & larger variation do not correlate better with their mature miRNAs (Table S10).

Comment 6: Does including genes that show less correlation than the threshold used (r=0.75) between CAGE & RNAseq increase noise in the model? Can the performance change be measured and discussed in the text?

Response: Thanks for the great comments! We chose this threshold (r=0.75) based on the false discovery cutoff 1. Because we want the trained model is applicable to both RNA-seq and CAGE data, it is necessary to use only those mRNAs that show consistent expression between the two platforms. With this said, including mRNAs with smaller correlation may introduce uncertainty to the model and prevent its application to the RNA-seq data or CAGE data. 

Comment 7: finally I would encourage the authors to publish their model in a publicly accessible repository.

Response: Thanks for the suggestion. We provided the full model at https://doi.org/10.6084/m9.figshare.21578847.v1.

Reviewer #2: 

Comment 1: In Table S4, the miRNA nomenclature is non standard. the "star" (*) notation to denote passenger strands has been abandoned a long time ago and mature miRNAs are named after the portion (5p, 3p) of the corresponding hairpin they arise from. Furthermore, the targetScan_targets field is "N/A" for ~235 out of 278 miRNAs. However, most of these miRNAs "without targetScan tqargets" have a long list of targets predicted by targetScan. I suspect that automatic retrieval of targetScan targets failed (perhaps due to inappropriate nomenclature of mature miRNAs in the dataset by the authors). This casts a serious shadow on the claim that "These associated mRNAs differ from their corresponding target mRNAs" as in 85% of the cases the miRNAs targets are (mistakingly) missing, hence they of course do not overlap "associated genes". This part of the analysis should be repeated after fixing the issue with the missing targetScan targets.

Response: Thanks for the great comments! Sorry for the confusion about the miRNA star here. We would like to point out that the "star" (*) notation in our table is not used to denote the passenger strands, which should be following the miRNA names. Instead, we added the "star" (*) in front of the miRNA names to indicate the name of miRNAs in mirGeneDB previously and now removed them. Thanks for pointing out the issue with the TargetScan targets. We previously used the targets with the conserved sites and now we used both conserved and non-conserved sites by TargetScan to define the targets. Our conclusions still hold. 

Comment 2: Figure quality can be dramatically improved in terms of resolution, labeling, clarity of legends, and, most importantly, the kind of data reported. 

In Figure 1 in its current form the only take home message is that correlations assessed by Pearson or Spearman algorithms yield comparable values. However, the point here is how good are the observed correlations (compared to a putative null hypothesis). Therefore, comparing the distribution of correlation values obtained using LASSO modeling with mock data using a boxplot or violinplot would be more appropriate.

It is not entirely clear to me what is depicted in Fig 1A, did the authors compare the "real" values and "computed" values on the very same identical data (i.e. traning data == test data)? If this is the case, such validation is tautological and should be removed. Furthermore, what is the rationale for choosing 10-fold cross validation (over 5-fold cross validation or any other n-fold?). The number of samples (> 1800) is very large, a robust trend should be validated by simply splitting the dataset in two halves. n-fold cross validation is usually more appropriate when dataset size is limited.

In figure 2 only 2 example miRNAs are reported. A cumulative picture reporting the general trend for all miRNAs should be provided instead. Furthermore, mirTarBase and targetscan targets should be analysed separately. Alternatively a boxplot displaying side to side (1) the distribution of the median correlation with "associate genes" for each miRNA (2) the distribution of the median correlation with targetScan target genes for each miRNA (3) the distribution of the median correlation with mirTarBase target genes for each miRNA.

Figure 3 Rather than reporting the observed outcome for each of the 10-fold cross validations, an average (or median) +/- SD should be reported. Howevver, check my comment above about the choice of 10-fold cross validation over other validation strategies available.

Response: Thanks for the constructive comments! We updated the figures. For figure 1, we replaced it with box plots of correlations from ten two-fold validation experiments as you suggested. We revised the validation method. The new validation method is ten experiments of two-fold validation. We randomly choose half of the samples in each experiment as training and the other half as testing. In addition, we removed the same primary cell types, cell lines, or tissues used in training from the testing samples in each experiment. What we compared is the predicted expression in the testing data with the known expression in the testing data. We assessed the significance based on the asymptotic distribution of the correlation (t-distribution) as we have enough testing samples here. For figure 2, we used a violin plot to display three distributions side by as you suggested. We revised figure 3 due to the usage of the two-fold cross validation here. We also added additional description related to figure 3 in the manuscript. 

Comment 3: The cutoff explored by the authors (a value > 0 in at least 20% of samples) for miRNA-TSS CAGE data will yield a zero inflated dataset. This is a relevant limitation of the study. Having a 0 in 80% of cases implies that any algorithm predicting always "0" will attain an 80% success. The authors should consider a more stringent cutoff to avoid such issues, restricting their analysis to the pri-miRNAs which exhibit a value > 0 in 80% of the cases. Otherwise, a strategy to cope with zero inflated data should be outlined and applied. Incidentally, 80% > 0 is the cutoff the authors apply to mature miRNA analysis. One might wonder why two such different cutoffs (20% > 0 for pri-miRNA and 80% > 0 for mature miRNAs have been applied in the same manuscript.

Response: Thanks for the great comments! We used the cutoff 80% in the entire manuscript now. 

Comment 4: the claim that data reported in ref (31) are due to an error or misinterpretaion of the data by the authors i not supported by the data. While I totally agree with Li and colleagues as long as the point is that pri-miRNAs and the corresponding mature miRNAs display a rather poor correlation (and I believe this makes pri-miRNA quantification a very poor proxy for mature miRNA expression) in (31) Fig 3C de Rie et al have compared the distribution of correlations observed between CAGE-pri-miRNA data (median ~ 0.2 as assessed by the figure) with the ones observed after breaking ties in the data (randomly permuting CAGE-pri-miRNA data and corresponding mature miRNA data), an accepted method of mimicking the null hypothesis. de Rie et al make a formally correct statement that the amount of correlation they observe is significantly higher than the one expected by chance. Furthermore the numbers they observed match quite closely the ones reported by Li inhere (a 55 percentile of 0.201).

So de Rie et al are correctly reporting a statistically significant correlation between CAGE-pri-miRNA and mature miRNAs, and their data perfectly agree with the analysis reported inhere. If the authors feel that a median correlation of 0.2 (although statistically significant) may be of rather poor interest from a biological point of view, the discussio would be the right session of the manuscript to laborate on that, but a negative comment on de Rie et al in the results section (while reporting essentially the same numbers) to question their interptretation of the data is inappropriate.

I would also state here that I am not in any way associated with any of the authors of (31).

Response: 

Response: Thanks for the great comments and unbiased assessment! We removed the negative comment on de Rie et al. We agree with your comments on the significance of the median correlation by de Rie et al. and the fact that there is no contradiction with our analysis. What we focused here is the individual miRNAs instead of the median correlation of all miRNAs. We measured the significance of the expression correlation of a pri-miRNA and its mature miRNA based on the asymptotic distribution and found that the majority of pri-miRNAs do not have a correlated expression with their mature miRNAs in the modified version. But we did remove the negative comments on de Rie et al.’s work.

Comment 5: The GO terms analysis is not reported, not even in the supplementary data. It would be interesting to see which are the GO terms associated with "associated genes" of each miRNA, whether they match the GO terms associated with their target (e.g. it would be conceivable that a set of regulators or transcription factors involved in apoptosis affect (and are hence associated with) the transcription of a mature miRNA targeting apoptosis related genes). Th authors should provide the full output of their GO terms analysis and elaborate on that.

Response: Thank you for pointing this out. We include Table S4 for GO terms associated with associated genes. We also have further discussions about the GO term results. 

Comment 6: page 14 "or the three neighborhood sizes, the median number of associated mRNAs identified was 164, 163, and 164, respectively, indicating the robustness of the inferred associated mRNAs" This is not correct. The fact that using different neighborhood sizes does not affect the associated mRNAs is rather telling us that the neighborhood window has a poor effect on the assessment of pri-miRNA-CAGE, thus yielding consistent input data (regardless of the neighborhood size) to the same LASSO algorithm, which obviously results into very similar output. However, any conclusion on the robustness of the algorithm itself would be an over-interpretaion of this observation. The robustness of the approach could be validated instead by testing it independently on different datasets (or different slices of the same dataset).

Response: Thanks for the comment! We agree that what you point out could be an alternative explanation of the similar median number of associated mRNAs. As there is no much additional independent samples for testing, we removed the robustness statement. 

Comment 7: In table S4 some miRNAs are repeated twice (e.g miR-101-1, miR-29b-2) and in some cases (miR-101-1) the TSS taken into account are almost completely overlapping. 

Response: Thanks for pointing it out. Some miRNAs TSS are reported from different studies, and some of them are very close to each other. We would like to show if there are any differences in associate genes as the result of these TSS differences. After we focus our study with miRNAs expressed in 80% of the samples, only hsa-mir-374a, hsa-mir-1303, hsa-mir-545 have multiple occurrence. We found that different TSSs of these miRNAs do have different associated mRNAs. The table is now renamed as Table S5. 

Comment 8: the datasets, miRNAs and samples analysed, although described in the text, should be described in tables depicting the number of samples, miRNAs and pri-miRNA CAGE fed into each analysis. GO terms output should also be provided in tables (and possibly representative examples reported as figures). The validation data about the LASSO approach should also be reported as supplementary tables (i.e. values underlying Fig 1).

Response: Thanks for the suggestions. We updated the supplementary data. The number of samples is indicated in supplementary file names. The GO terms output is now Table S4 and adding Figure 4 as representative examples. We attached the original data used for the previous figure 1 in Tables S6 and S7. 

Comment 9: the authors should elaborate on the biological significance of inferring pri-miRNA expression based on a set of mRNAs. Why this finding is potentially interesting from a biological point of view? do they expect that "associate mRNAs" may encode for transcription factors, RNA binding enzymes or other proteins directly involved in promoting/repressing transcription of pri-miRNAs?

Response: Thanks for the suggestion. The significance to infer the pri-miRNA expression is that we cannot measure the pri-miRNA expression easily. But we can measure the expression of mRNAs. If we can model it through the expression of mRNA expression, it may provide a way to study pri-miRNA expression directly, especially in the RNA-seq experiments. We emphasized this point in the introduction and discussion. About the function of the associated mRNAs, we analyzed their function in Table S4.

---

## [Decision Letter · Decision Letter 1]

12 Dec 2022

PONE-D-22-18168R1A computational modeling of pri-miRNA expressionPLOS ONE

Dear Dr. Li,

Thank you for submitting your manuscript to PLOS ONE. After careful consideration, we feel that it has merit but does not fully meet PLOS ONE’s publication criteria as it currently stands. Therefore, we invite you to submit a revised version of the manuscript that addresses the points raised during the review process.

It is mandatory that you address the points raised by Reviewer 2, regarding the meaning and significance of the results described. Please submit your revised manuscript by Jan 26 2023 11:59PM. If you will need more time than this to complete your revisions, please reply to this message or contact the journal office at plosone@plos.org. Please include the following items when submitting your revised manuscript:A rebuttal letter that responds to each point raised by the academic editor and reviewer(s). You should upload this letter as a separate file labeled 'Response to Reviewers'.A marked-up copy of your manuscript that highlights changes made to the original version. You should upload this as a separate file labeled 'Revised Manuscript with Track Changes'.An unmarked version of your revised paper without tracked changes. You should upload this as a separate file labeled 'Manuscript'.

We look forward to receiving your revised manuscript.

Kind regards,

Manuela Helmer-Citterich

Academic Editor

PLOS ONE

Reviewers' comments:

Reviewer's Responses to Questions

**Comments to the Author**

1. If the authors have adequately addressed your comments raised in a previous round of review and you feel that this manuscript is now acceptable for publication, you may indicate that here to bypass the “Comments to the Author” section, enter your conflict of interest statement in the “Confidential to Editor” section, and submit your "Accept" recommendation.

Reviewer #1: (No Response)

Reviewer #2: (No Response)

2. Is the manuscript technically sound, and do the data support the conclusions?

Reviewer #1: Yes

Reviewer #2: No

3. Has the statistical analysis been performed appropriately and rigorously? 

Reviewer #1: Yes

Reviewer #2: No

4. Have the authors made all data underlying the findings in their manuscript fully available?

Reviewer #1: Yes

Reviewer #2: Yes

5. Is the manuscript presented in an intelligible fashion and written in standard English?

Reviewer #1: Yes

Reviewer #2: Yes

6. Review Comments to the Author

Reviewer #1: Thanks to the authors for their replies & modifications to the text. Below my comments:

- Figures resolution should be highly improved.

- In Figure 2, I noticed that some of the associated mRNAs show correlation of 1 with the pri-miRNA. Can the authors resolve the potential concern that these mRNAs are identical to the modeled pri-miRNAs? If removed, how would this affect the performance of the model? This raises potential concerns of contamination between input and predicted variables, that if confirmed, the authors should address by comparing the distribution of performance across pri-miRNA with the performance after removing those putatively identical data points to the predicted values.

Reviewer #2: The authors have only partially addressed my comments. Overall, conclusions are not supported by the data as key controls are still lacking.

In particular, Fig 1, which underlies tha main finding of the paper is still lacking a "negative control" (I had suggested comparing correlations observed using the LASSO model to some kind of mock data, Comment #2 of my review).

Furthermore, Fig 2 is now reporting essentially identical plots for "associate" vs "target" mRNAs, with the notable exception of a peak centered on Correlation =1. However the statistical (and biological) significance of Pearson (or Spearman) values > 1 (clearly depicted in Fig2) is questionable... how could the authors possibly obtain values > 1? and if they did not, which kind of "smoothing" algorithm is resulting in such an artifact?

The data reported in Table S7 (which does not match the description reported in the manuscript) report identical data regardless of the use of De Rie et al data or the author's data. Again, this further reinforces the opinion by this reviewer that any divergence between De Rie et al and the authors of this manuscript pertains the interpretation of the data, not the results. Therefore the Results paragraph dedicated to this issue is pointless.

The GO term analysis relies on FDR <1. This cutoff is meaningless, as only excludes tests yielding a FDR=1 while retaining anything with an FDR = 0.9 or 0.8 which are obviously unreliable findings. Furthermore the FDR for each test is not reported (only P-values are available). Therefore, no conclusion can be drawn on this analysis.

7. PLOS authors have the option to publish the peer review history of their article (what does this mean?). If published, this will include your full peer review and any attached files.

Reviewer #1: No

Reviewer #2: No

---

## [Author Response · Author response to Decision Letter 1]

17 Dec 2022

Reviewer #1

Comment 1: Thanks to the authors for their replies & modifications to the text. Below my comments:

- Figures resolution should be highly improved.

Response: Thanks for the comments! We updated the figures to increase the font sizes and resolution. The figures especially fig, 4 may still look vague in the pdf because of the plos one online submission conversion. But if you click the blue links on the top of the pdf pages that contain the figures, you can see the high resolution figures. 

Comment 2: In Figure 2, I noticed that some of the associated mRNAs show correlation of 1 with the pri-miRNA. Can the authors resolve the potential concern that these mRNAs are identical to the modeled pri-miRNAs? If removed, how would this affect the performance of the model? This raises potential concerns of contamination between input and predicted variables, that if confirmed, the authors should address by comparing the distribution of performance across pri-miRNA with the performance after removing those putatively identical data points to the predicted values.

Response: Thanks for the great comments! Among the 195 pri-miRNAs we considered in this study, there are 3 pri-miRNAs with their Pearson’s correlation coefficient equal to 1 with one and only one of their associated mRNAs. In total, there are 14 pri-miRNAs with one and only one associated mRNA having their TSSs within +/-100 base pairs of the corresponding pri-miRNA TSS. We removed these associated mRNAs and retrained the model when we considered the expression of these pri-miRNAs. We found that the minimum, mean, and median Pearson’s correlation of the predicted and true expression per miRNA was 0.78, 0.91 and 0.92, respectively. The minimum, mean and median Spearman’s correlation per miRNA was 0.27, 0.82 and 0.83, respectively. When measured the similarity of the predicted expression in every sample, the minimum, mean, and median Pearson’s correlation per sample was 0.45, 0.87 and 0.90, respectively. The minimum, mean and median Spearman’s correlation per sample was 0.39, 0.83 and 0.85, respectively. Based on the almost identical correlation to the correlation with the original model we presented, it is safe to say the mRNAs with similar TSS with pri-miRNA should not be a concern in this study. We modified the paper correspondingly.

Reviewer #2

Comment 1: The authors have only partially addressed my comments. Overall, conclusions are not supported by the data as key controls are still lacking.

- In particular, Fig 1, which underlies tha main finding of the paper is still lacking a "negative control" (I had suggested comparing correlations observed using the LASSO model to some kind of mock data, Comment #2 of my review).

Response: Thanks for the feedback. Sorry that we thought we addressed this issue with the asymptotic distribution of the correlation and did not use the negative controls previously. We now added two different negative controls. For the negative control #1, we randomly permute each mRNA expression across the 1829 samples while keep the same pri-miRNA expression to see how well the pri-miRNA expression can be modeled by these randomized mRNA expression data. For the negative control #2, we randomly assign a mRNA expression by permuting the expression of all mRNAs in one sample so that overall expression of a mRNA across the 1829 samples becomes random. We then ran the same analysis to model the pri-miRNA expression. We compared the correlation of the predicted pri-miRNA expression and the true pri-miRNA expression from the two negative control datasets with the correlation obtained from the original data in the manuscript. See the comparison in the following figure (with a model based on all 1829 samples) and figure 1 (with a model based on the two-fold cross-validation) in the paper. The correlation from the original data is much larger than that from the negative control data.

Comment 2: Furthermore, Fig 2 is now reporting essentially identical plots for "associate" vs "target" mRNAs, with the notable exception of a peak centered on Correlation =1. However the statistical (and biological) significance of Pearson (or Spearman) values > 1 (clearly depicted in Fig2) is questionable... how could the authors possibly obtain values > 1? and if they did not, which kind of "smoothing" algorithm is resulting in such an artifact?

Response: Thanks for pointing out this mistake. The correlation value was not larger than 1. We used the Python Seaborn package to plot the violin plot, which features a kernel density estimation of the underlying distribution. This is the reason why the distribution seems >1 but there’s actually no correlation >1. We have redrawn the figure 2 in the current version. We also removed the mRNAs with their TSSs within +/-100 base pairs of the pri-miRNA TSS when modeling the expression of this pri-miRNA. Our conclusions still hold. 

Comment 3: The data reported in Table S7 (which does not match the description reported in the manuscript) report identical data regardless of the use of De Rie et al data or the author's data. Again, this further reinforces the opinion by this reviewer that any divergence between De Rie et al and the authors of this manuscript pertains the interpretation of the data, not the results. Therefore the Results paragraph dedicated to this issue is pointless.

Response: Thanks for the comments! We are not sure we precisely understand your points here. Table S7 is about how well the predicted pri-miRNA expression correlates with the actual pri-miRNA expression per sample, which has no relation with De Rie et al.’s point that the expression of pri-miRNAs and mature miRNAs correlates well. In other words, Table S7 is about the modelling of pri-miRNA expression with mRNAs, which does not deal with mature miRNA expression. De Rie et al. did not model pri-miRNA expression in their study. In terms of the results, the modeling of pri-miRNA expression by mRNA expression is not in De Rie et al.’s study, which is the main purpose of our study. Moreover, about pri-miRNA and mature miRNA expression correlation, our results are different from De Rie et al.’s as well. We showed that the majority pri-miRNAs do not have correlated expression with their mature miRNAs for individual miRNAs (FDR<0.01), which is supported by our data and De. Rie et al.’s data. De Rie et al. showed the AVERAGE expression correlation between pri-miRNAs and their mature miRNAs is significant and concluded that the pri-miRNAs have a correlated expression with their mature miRNAs. In fact, more than half of the pri-miRNAs do not correlate well with their mature miRNAs (Supplementary Table S9). 

Comment 4: The GO term analysis relies on FDR <1. This cutoff is meaningless, as only excludes tests yielding a FDR=1 while retaining anything with an FDR = 0.9 or 0.8 which are obviously unreliable findings. Furthermore the FDR for each test is not reported (only P-values are available). Therefore, no conclusion can be drawn on this analysis.

Response: Thank you for pointing this out! We agree that the FDR cutoff should be smaller. We have changed the FDR cutoff to 0.1. We also report the FDR together with p-values.

---

## [Decision Letter · Decision Letter 2]

24 Feb 2023

PONE-D-22-18168R2A computational modeling of pri-miRNA expressionPLOS ONE

Dear Dr. Li,

Thank you for submitting your manuscript to PLOS ONE. After careful consideration, we feel that it has merit but does not fully meet PLOS ONE’s publication criteria as it currently stands. Therefore, we invite you to submit a revised version of the manuscript that addresses the points raised during the review process. Both reviewers raised issues about the statistics regarding your results and not for the first time. I kindly ask you to give better answers with respect to the ones given in the last version of your manuscript. I am very sorry for the time that you as Authors, the Reviewers and myself are spending on this submission and I hope that you will be able to produce reliable data for the Reviewers concerns.

We look forward to receiving your revised manuscript.

Kind regards,

Manuela Helmer-Citterich

Academic Editor

PLOS ONE

Reviewers' comments:

Reviewer's Responses to Questions

**Comments to the Author**

1. If the authors have adequately addressed your comments raised in a previous round of review and you feel that this manuscript is now acceptable for publication, you may indicate that here to bypass the “Comments to the Author” section, enter your conflict of interest statement in the “Confidential to Editor” section, and submit your "Accept" recommendation.

Reviewer #2: (No Response)

2. Is the manuscript technically sound, and do the data support the conclusions?

Reviewer #2: No

3. Has the statistical analysis been performed appropriately and rigorously? 

Reviewer #2: No

4. Have the authors made all data underlying the findings in their manuscript fully available?

Reviewer #2: Yes

5. Is the manuscript presented in an intelligible fashion and written in standard English?

Reviewer #2: Yes

6. Review Comments to the Author

Reviewer #2: The authors have addressed comment 1, but several inconsistencies between data in supplementary Tables and claims in the text and figures remain.

Comment 1: The authors have fully addressed my comment.

Comment 2: The authors have simply cut the y axis at y=1. Obviously, values aove 1 disappeared. This is not acceptable. Furthermore, the copy of Fig 2 reported in the author's response is different from the one reported in the text, and both differ from the one which was in the previous version.

If values > 1 are not in the original data but are the result of a smoothing function (kernel density) this should be documented. Nevertheless, my original question still holds: what is the meaning od Pearson correlation =1 in this plot? How do the authors explain this extreme value? which pairs do yield such astonishingly high correlations? Does this extreme result stem from pri-miRNA:mRNA pairs which, perhaps due to duplicated annotations, actually represent the same RNA molecule (or two largely overlapping RNA moleculesl? This must be investigated picking those pairs and checking annotation, the simple claim that RNAs with TSS within 100 nt of the pri-miRNA TSS have been removed is not sufficient.

Comment 3; Table S7 is enitled "Table_S7_The_Pearsons_and_spearmans_correlation_coefficient_between_pri-iRNAs_and_mature_miRNAs". Furthermore, the reported values (column 1) have an average value of 0.17, which is at the odds with the correlation values reported in Fig 1. If Table S7 reports (as the authors wrote in their reply) how well the predicted pri-miRNA expression correlates with the actual pri-miRNA expression per sample (which are supposed to be the correlation values undelying Fig 1, blue boxplot, average~0.6), there is a clear inconsistency in the data, as the average value in Table S7 is 0.17. I therefore suspect that, in line with the actual title of the Table S7 and the column names (pearson_corr_based_on_our spearman_corr_based_on_our pearson_corr_based_on_derie spearman_corr_based_on_derie) this Table reports on the miRNA:pri-miRNA correlation using two different dataset (stemming actually from the same experiment) and again these data further confirm that miRNAs display a minor yet significant correlation with pri-miRNAs in both datasets.

Comment 4: In Table S4 FDR is not reported, only P-values are reported. Furthermore, while data in Table S4 have been computed for each single miRNA, Fig 4 does not report a specific miRNA, so it is unclear how those GO terms were obtained.

7. PLOS authors have the option to publish the peer review history of their article (what does this mean?). If published, this will include your full peer review and any attached files.

Reviewer #2: No

---

## [Author Response · Author response to Decision Letter 2]

25 May 2023

Reviewer #2: 

Comment 1: The authors have addressed comment 1, but several inconsistencies between data in supplementary Tables and claims in the text and figures remain.

The authors have fully addressed my comment.

Response: We appreciate your feedback and are glad we could address your concern related to comment 1. We will respond to the inconsistencies you mentioned below in the following sections.

Comment 2: The authors have simply cut the y axis at y=1. Obviously, values aove 1 disappeared. This is not acceptable. Furthermore, the copy of Fig 2 reported in the author's response is different from the one reported in the text, and both differ from the one which was in the previous version.

If values > 1 are not in the original data but are the result of a smoothing function (kernel density) this should be documented. Nevertheless, my original question still holds: what is the meaning od Pearson correlation =1 in this plot? How do the authors explain this extreme value? which pairs do yield such astonishingly high correlations? Does this extreme result stem from pri-miRNA:mRNA pairs which, perhaps due to duplicated annotations, actually represent the same RNA molecule (or two largely overlapping RNA moleculesl? This must be investigated picking those pairs and checking annotation, the simple claim that RNAs with TSS within 100 nt of the pri-miRNA TSS have been removed is not sufficient.

Response: Thank you for the feedback! 

Here is the reason that we cut the y-axis at y=1 in the plot. As we mentioned before, the kernel density estimation caused the initial figure in the first submitted version to seem to have values >1, although there were no value >1. To avoid this confusion, we adjusted the plot accordingly in the last revision. 

The correlation = 1 means that there is indeed an overlap of The TSS regions of primary miRNAs and their associated genes. As we mentioned in the previous response, among the seventeen miRNA-mRNA pairs with their TSSs close to 100 base pairs, three pairs have a correlation value 1 and five pairs have a correlation value close to 1. We removed these primary miRNA and mRNA pairs when we predicted the primary miRNA expression in the last version. However, we did include these pairs in other analyses in the last version, which resulted in certain pri-miRNA-mRNA pairs with correlation 1 in Fig. 2. To effectively address the issue caused by miRNA-mRNA pairs with close TSS regions, we removed all these pri-miRNA-mRNA pairs to perform the lasso analysis and updated the lasso model and the entire manuscript this time.

Regarding the figures, we want to emphasize that there is no inconsistency between the figures provided in the manuscript and our response. The figure we included in our previous response was based on two different negative control datasets you suggested, which we pointed out in the previous response letter. The Fig. 2 in the previous manuscript compares the expression correlation of the primary miRNAs and their associated genes with the expression correlation of the pri-miRNAs and their miTarBase and TargetScan target genes. The two figures are thus supposed to be different. We did not include the figure in the response in the manuscript because the Fig. 1 in the previous manuscript already described the same analysis with two negative control datasets in a different format. We apologize for any confusion that may have arisen and hope that this clarifies the matter.

Comment 3: Table S7 is enitled "Table_S7_The_Pearsons_and_spearmans_correlation_coefficient_between_pri-iRNAs_and_mature_miRNAs". Furthermore, the reported values (column 1) have an average value of 0.17, which is at the odds with the correlation values reported in Fig 1. If Table S7 reports (as the authors wrote in their reply) how well the predicted pri-miRNA expression correlates with the actual pri-miRNA expression per sample (which are supposed to be the correlation values undelying Fig 1, blue boxplot, average~0.6), there is a clear inconsistency in the data, as the average value in Table S7 is 0.17. I therefore suspect that, in line with the actual title of the Table S7 and the column names (pearson_corr_based_on_our spearman_corr_based_on_our pearson_corr_based_on_derie spearman_corr_based_on_derie) this Table reports on the miRNA:pri-miRNA correlation using two different dataset (stemming actually from the same experiment) and again these data further confirm that miRNAs display a minor yet significant correlation with pri-miRNAs in both datasets.

Response: Thank you for your comment. Regarding your concern about the relationship between Figure 1 and Table S9 (previously S7), we would like to clarify that they represent different analyses and are not directly comparable. Figure 1 shows the results of ten two-fold cross-validation experiments for predicting pri-miRNA expression, which calculated the correlation between the true expression and the predicted expression of pri-miRNAs, with comparison to two different negative control datasets. On the other hand, Table S9 (previously S7) shows the correlations between pri-miRNA expression and mature miRNA expression. Pri-miRNA expression is different from the mature miRNA expression. The two results are thus not directly connected and should not be compared against each other. Furthermore, we want to emphasize that our analysis in Table S9 shows that about half of the pri-miRNAs do not correlate well with their mature miRNAs. This highlights the importance of analyzing each individual pri-miRNA, rather than looking at the overall average.

Comment 4: In Table S4 FDR is not reported, only P-values are reported. Furthermore, while data in Table S4 have been computed for each single miRNA, Fig 4 does not report a specific miRNA, so it is unclear how those GO terms were obtained.

Response: Thank you for taking the time to review our work and provide feedback. We would like to respectfully clarify that the FDR q-value for each GO term was reported in Table S4. Moreover, we would like to emphasize that Fig 4 does indeed report the GO results based on the function of the associated genes of a specific miRNA, hsa-miR-92a, as stated in the figure legend.

---

## [Decision Letter · Decision Letter 3]

4 Jul 2023

PONE-D-22-18168R3A computational modeling of pri-miRNA expressionPLOS ONE

Dear Dr. Li,

Thank you for submitting your manuscript to PLOS ONE. After careful consideration, we feel that it has merit but does not fully meet PLOS ONE’s publication criteria as it currently stands. Therefore, we invite you to submit a revised version of the manuscript that addresses the points raised during the review process.

We look forward to receiving your revised manuscript.

Kind regards,

Manuela Helmer-Citterich

Academic Editor

PLOS ONE

Journal Requirements:

**Additional Editor Comments:**

The Authors are kindly asked to follow the last suggestions of the reviewer and make the requested modifications.

Reviewers' comments:

Reviewer's Responses to Questions

**Comments to the Author**

1. If the authors have adequately addressed your comments raised in a previous round of review and you feel that this manuscript is now acceptable for publication, you may indicate that here to bypass the “Comments to the Author” section, enter your conflict of interest statement in the “Confidential to Editor” section, and submit your "Accept" recommendation.

Reviewer #2: (No Response)

2. Is the manuscript technically sound, and do the data support the conclusions?

Reviewer #2: Yes

3. Has the statistical analysis been performed appropriately and rigorously? 

Reviewer #2: Yes

4. Have the authors made all data underlying the findings in their manuscript fully available?

Reviewer #2: Yes

5. Is the manuscript presented in an intelligible fashion and written in standard English?

Reviewer #2: Yes

6. Review Comments to the Author

Reviewer #2: In these third revision as well as in the second revision, although the text reported for the figshare link for Supplementary tables has been updated ("21578847.v5"), the underlying hypertext link is still pointing to v1 (https://figshare.com/articles/online_resource/A_computational_modeling_of_pri-miRNA_expression/21578847/1), for this reason I could not access the appropriate version of the Supplementary Tables during second round of revision, thus explaining why I saw inconsistencies between the Tables published on figshare and the text.

The link should be updated.

I still have concerns regarding some of the statements in section "The expression of the majority of mature miRNAs do not correlate

well with the expression of pri-miRNAs", for example:

"with a 57.71 percentile of 0.1592" is a really odd statement, why should one actually choose the "57.71th percentile" of a distribution? Does this value perform any better than the median (50 percentile) which is commonly used instead?

Finally, the authors should acknowledge somewhere that the values reported in Table S9 show that correlation coefficients computed here and in de rie are in fact identical in 154/176 miRNAs, and when they differ, in most cases the difference is negligible (e.g. miR-23a: 0.609994705145189 vs 0.609835708540695) so the point here is entirely on data interpretation, not on the data.

7. PLOS authors have the option to publish the peer review history of their article (what does this mean?). If published, this will include your full peer review and any attached files.

Reviewer #2: No

---

## [Author Response · Author response to Decision Letter 3]

21 Jul 2023

Comment 1: In these third revision as well as in the second revision, although the text reported for the figshare link for Supplementary tables has been updated ("21578847.v5"), the underlying hypertext link is still pointing to v1 (https://figshare.com/articles/online_resource/A_computational_modeling_of_pri-miRNA_expression/21578847/1), for this reason I could not access the appropriate version of the Supplementary Tables during second round of revision, thus explaining why I saw inconsistencies between the Tables published on figshare and the text. The link should be updated.

Response: Thank you so much for helping us to correct the links. Now they are updated.

Comment 2: I still have concerns regarding some of the statements in section "The expression of the majority of mature miRNAs do not correlate

well with the expression of pri-miRNAs", for example:

"with a 57.71 percentile of 0.1592" is a really odd statement, why should one actually choose the "57.71th percentile" of a distribution? Does this value perform any better than the median (50 percentile) which is commonly used instead?

Response: Thank you for the feedback! As we stated in the paper, it is the cutoff with the FDR of 0.01. We also added this information at another place in the text in the modified version.

Comment 3: Finally, the authors should acknowledge somewhere that the values reported in Table S9 show that correlation coefficients computed here and in de rie are in fact identical in 154/176 miRNAs, and when they differ, in most cases the difference is negligible (e.g. miR-23a: 0.609994705145189 vs 0.609835708540695) so the point here is entirely on data interpretation, not on the data.

Response: Thank you for your comment. We added two sentences in the text to acknowledge that these correlation coefficients are quite similar to those in the previous studies, and the difference of our conclusion is based on different aspects of consideration

---

## [Decision Letter · Decision Letter 4]

26 Jul 2023

PONE-D-22-18168R4A computational modeling of pri-miRNA expressionPLOS ONE

Dear Dr. Li,

Thank you for submitting your manuscript to PLOS ONE. After careful consideration, we feel that it has merit but does not fully meet PLOS ONE’s publication criteria as it currently stands. Therefore, we invite you to submit a revised version of the manuscript that addresses the points raised during the review process.

We look forward to receiving your revised manuscript.

Kind regards,

Manuela Helmer-Citterich

Academic Editor

PLOS ONE

Journal Requirements:

**Additional Editor Comments:**

After so many revisions, I ask the Authors to consider the comments of the Reviewer to their last revision and make the requested amendments.

Reviewers' comments:

Reviewer's Responses to Questions

**Comments to the Author**

1. If the authors have adequately addressed your comments raised in a previous round of review and you feel that this manuscript is now acceptable for publication, you may indicate that here to bypass the “Comments to the Author” section, enter your conflict of interest statement in the “Confidential to Editor” section, and submit your "Accept" recommendation.

Reviewer #2: (No Response)

2. Is the manuscript technically sound, and do the data support the conclusions?

Reviewer #2: Partly

3. Has the statistical analysis been performed appropriately and rigorously? 

Reviewer #2: No

4. Have the authors made all data underlying the findings in their manuscript fully available?

Reviewer #2: Yes

5. Is the manuscript presented in an intelligible fashion and written in standard English?

Reviewer #2: Yes

6. Review Comments to the Author

Reviewer #2: Comment 2: I still have concerns regarding some of the statements in section "The expression of the majority of mature miRNAs do not correlate well with the expression of pri-miRNAs", for example: "with a 57.71 percentile of 0.1592" is a really odd statement, why should one actually choose the "57.71th percentile" of a distribution? Does this value perform any better than the median (50 percentile) which is commonly used instead?

> Response: Thank you for the feedback! As we stated in the paper, it is the cutoff with the FDR of 0.01. We also added this information at another place in the text in the modified version.

The methods section does not provide details on how the "FDR of 0.01" was calculated. This should be explained clearly. It is also unclear the statement "with a 57.71 percentile of 0.1592 (FDR<0.01)", the FDR reported is linked to a test about the likelyhood that the 57.71th percentile is 0.1592 or is simply the FDR associated with the corrlation observed for the pri-miRNA ranking at the 57.71th percentile.

7. PLOS authors have the option to publish the peer review history of their article (what does this mean?). If published, this will include your full peer review and any attached files.

Reviewer #2: No

---

## [Author Response · Author response to Decision Letter 4]

14 Aug 2023

Editor Comments: 

Comment 1: After so many revisions, I ask the Authors to consider the comments of the Reviewer to their last revision and make the requested amendments.

Response: Thanks for the suggestion. Yes, we revisited the reviewer comments from the last revision and provided the requested details in the modified version.

Reviewer #2: 

Comment 1: The methods section does not provide details on how the "FDR of 0.01" was calculated. This should be explained clearly. It is also unclear the statement "with a 57.71 percentile of 0.1592 (FDR<0.01)", the FDR reported is linked to a test about the likelyhood that the 57.71th percentile is 0.1592 or is simply the FDR associated with the corrlation observed for the pri-miRNA ranking at the 57.71th percentile. 

Response: Thanks for the feedback! The FDR was calculated with the well-known Benjamini Hochberg algorithm. In brief, we calculated the p-value for the correlation of the pri-miRNA expression and its mature miRNA expression based on the asymptotic distribution of the correlation for each pri-miRNA. We then ranked the pri-miRNAs based on their p-values from the smallest p-value to the largest p-value. Finally, we checked whether the sum of the p-values from the first pri-miRNA to the current i-th one was larger than 0.01 for i starting from 1 and stopped when the first sum was larger than 0.01. When we stopped at this specific i-th pri-miRNA, we reported the top (i-1) miRNAs as significantly correlated pri-miRNAs with FDR smaller than 0.01. In our case, this i-th miRNA had a correlation of 0.1592 (the larger the correlation, the smaller the p-value), which corresponding to the 57.71-th percentile of the correlations. We modified the paper to include a brief description of this standard procedure.

---

## [Editor Report · Decision Letter 5]

16 Aug 2023

A computational modeling of pri-miRNA expression

PONE-D-22-18168R5

Dear Dr. Li,

We’re pleased to inform you that your manuscript has been judged scientifically suitable for publication and will be formally accepted for publication once it meets all outstanding technical requirements.

Kind regards,

Manuela Helmer-Citterich

Academic Editor

PLOS ONE
---

## [Editor Report · Acceptance letter]

21 Aug 2023

PONE-D-22-18168R5 

A computational modeling of pri-miRNA expression 

Dear Dr. Li:

I'm pleased to inform you that your manuscript has been deemed suitable for publication in PLOS ONE. Congratulations! Your manuscript is now with our production department. 

Kind regards, 

on behalf of

Dr. Manuela Helmer-Citterich 

Academic Editor

PLOS ONE